materials science/nanotechnology/energy

π-conjugated polymers, metal-ion batteries, redox activity, supercapacitors

**Authors for correspondence:**
Ashwani Kumar
e-mail: 01ashraj@gmail.com
Yogesh Kumar
e-mail: ykumar@arsd.du.ac.in

# A review of π-conjugated polymer-based nanocomposites for metal-ion batteries and supercapacitors

Santosh J. Uke[1], Satish P. Mardikar[2], Ashwani Kumar[3], Yogesh Kumar[4], Meenal Gupta[5] and Yogesh Kumar[6]

[1]Department of Physics, JDPS College, SGB Amravati University, Amravati India
[2]Department of Chemistry, SRS College, SGB Amravati University, Amravati India
[3]Institute Instrumentation Centre, IIT Roorkee-247667, India
[4]Department of Physics G.D, Goenka University, Gurgaon 122002, India
[5]Department of Physics, MRL, SBSR, Sharda University, Greater Noida 201 310, India
[6]Department of Physics, ARSD College, University of Delhi 110021, India

yk, 0000-0002-5567-4884; YK, 0000-0003-3535-2068

Owing to their extraordinary properties of π-conjugated polymers (π-CPs), such as light weight, structural versatility, ease of synthesis and environmentally friendly nature, they have attracted considerable attention as electrode material for metal-ion batteries (MIBs) and supercapacitors (SCPs). Recently, researchers have focused on developing nanostructured π-CPs and their composites with metal oxides and carbon-based materials to enhance the energy density and capacitive performance of MIBs and SCPs. Also, the researchers recently demonstrated various novel strategies to combine high electrical conductivity and high redox activity of different π-CPs. To reflect this fact, the present review investigates the current advancements in the synthesis of nanostructured π-CPs and their composites. Further, this review explores the recent development in different methods for the fabrication and design of π-CPs electrodes for MIBs and SCPs. In review, finally, the future prospects and challenges of π-CPs as an electrode materials for strategies for MIBs and SCPs are also presented.

This article has been edited by the Royal Society of Chemistry, including the commissioning, peer review process and editorial aspects up to the point of acceptance.

# 1. Introduction

The ever increasing energy demands and depletion of fossil fuel highlighted the requirements of energy production from renewable energy resources. For the industrial development and wellness of human beings, energy storage is of equal importance as energy production. To meet the future challenges of energy storage of the portable and flexible cutting-edge electronics, the world demands

high energy density, low cost, environmentally friendly and sustainable electrical energy storage system (ESS) [1–4]. Presently, in portable electronics batteries and supercapacitors (SCPs) are the primary choices as power sources to consumers. The lead-acid battery and lithium-ion battery (LIB) are fully commercialized. Both of these batteries have their own weaknesses and strengths. Specifically, the use of hazardous and banned lead as electrode material and lower volumetric energy density of lead-acid battery, it couldn't be the practical choice as an energy storage device for future sustainable energy management [5–7]. Therefore, at present, most of the portable and off-grid electrical devices solely rely on LIBs. In addition, the nickel-cadmium (Ni-Cd), nickel metal hydride (NI-MH), zinc-bromine (Zb-Br), vanadium redox (VR), sodium-sulfur (Na-S), nickel-hydrogen (Ni-H), nickel-zinc (Ni-Zn), molten salt, metal-air batteries etc. are the available emerging electrical energy storage device for future high-end electrical applications [8]. Furthermore, the SCPs are also promising and mature electrochemical energy storage technologies owing to their outstanding qualities such as low cost, moderate energy density, high specific capacitance, high retention in capacitance and environmentally friendly nature [9]. For commercialization of any electrical energy system, the parameters like energy density, power density, capacitance, cycle life and shelf life are the key parameters which further drive the effectiveness of these ESS [3,10,11]. The positive electrode (cathode), negative electrode (anode), electrolyte and separator are the main components of the ESS. More specifically, the output parameters of the ESS strongly depend on the types of electrodes and electrolytes employed in it. In the case of LIBs, the lithium contains metal oxide used and cathode and graphite as an anode. The lithium cobalt oxide (LiCoO), lithium manganese oxide (LiMn$_2$O$_4$), lithium iron phosphate (FeLiO$_4$P), lithium nickel cobalt aluminium oxide (LiNiCoAlO$_2$), lithium nickel manganese cobalt oxide (LiNiMnCoO$_2$) etc. have been used as positive electrode material for LIBs [12–14].

Recently, the various organic materials have attracted much more attention as electrode material for various energy storage applications. The outstanding properties of organic polymer include ease of synthesis, low cost of production, light weight, and environmental capability, processability and moulding ability etc. [15–18]. The carbonyl derivatives such as quinones, carboxylates, anhydrides, imides, and ketones, conductive polymers, organosulfur compounds, etc. have been widely used as organic electrode material for energy storage applications [17,18]. Out of the different organic polymers, the use of π-conjugated polymers (π-CPs) for energy storage applications has increased enormously. Interestingly, the π-CPs have many attracting properties towards energy storage applications, which include high electrical conductivity, cost-effective, light weight and environmentally friendly in nature [19,20]. Moreover, due to intrinsic properties of the π-CPs, they are highly flexible in nature, which imparts the flexibility of electrodes used in ESS.

Due to the high polymeric (long chain) nature and low cost of the π-CPs, has a very high demand as electrode material in flexible ESS. Furthermore, such flexible ESS are used in wearable devices such as light weight and high-energy density ESSs in flexible electronics, laptops, mobile phones, etc. The π-CPs such as PANI (polyaniline), PPy (polypyrrole), PTh (polythiophene), PEDOT (poly(3,4-ethylenedioxythiophene)), PPP (polyparaphenylene) and PI (polyindole) are the highly explored and potential electrode material for both the batteries and SCPs. The environmentally friendly nature and excellent electrical conductivity compared to other conducting polymers make π-CPs as sustainable electrode material for versatile ESS [21].

At present, there are many reviews in literature which show the recent progress of organic material as active electrode material for energy storage application. Many of them are focusing on the progress and applications of groups of materials such as organic materials, metal oxides and carbon-based materials as active electrode material for energy storage [22–25]. In this context, to enlighten the recent progress and advancement in electrode materials, in the present review, we comprehensively reviewed the special categories of organic material, i.e. π-CPs as electrode material for versatile ESS. Initially, we briefly introduced the various π-CPs followed by their fundamental intrinsic properties toward energy storage applications. Furthermore, we have focused on synthesis and recent progress in different synthesis methodologies of π-CPs. Additionally, we have summarized the applications of π-CPs towards versatile ESS as batteries and SCPs. The scope and challenges of π-CPs as electrode materials are presented. The present review could provide the detailed insight and future prospects toward the π-CPs as high-energy density and low-cost electrode material for ESS.

# 2. Methods of synthesis of π-conjugated polymers

For fabrication of conjugated structures in polymers and determining the final characteristics of electrode material in the ESS, the synthesis strategies play a crucial role. There are various synthetic strategies for

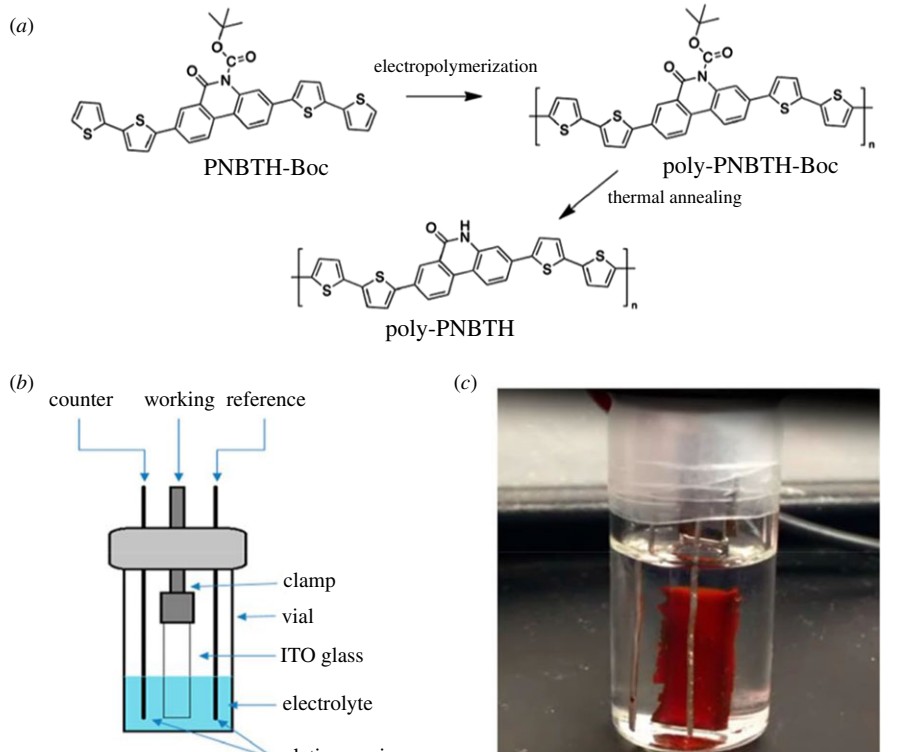

**Figure 1.** (*a*) Schematics of electro-polymerization of poly-PNBTH-Boc from monomer PNBTH-Boc, (*b*) three-electrode electro-polymerization reactor and (*c*) photo of poly-PBNTH-Boc film on the ITO electrode [43]. Images reproduced with permission.

synthesis of π-CPs, which includes chemical oxidation [26], photochemical method [27], inclusion method [28], metathesis method [29], emulsion (mini and micro) method [29,30], solid-state method [29,31], plasma polymerization [29,32], electro-polymerization [33] and copolymerization method [34]. Out of these synthetic strategies, the oxidative polymerization method of synthesis is a highly preferred strategy for the synthesis of π-CPs in bulk form [29], drop-casting method and spin coating method, etc. The synthesis strategies of different π-CPs such as pure π-CPs, derivatives of π-CPs and nanocomposites of π-CPs have been explained in subsequent sections.

The highly implemented and reported synthesis method for synthesis of pure π-CPs is oxidative polymerization. Many researchers have reported successful synthesis of π-CPs PANI, PPy, PTh, PI, (PEDOT), etc. via oxidative polymerization. The chemical oxidative polymerization method is a simple, clean and cost-effective method for synthesis of pure π-CPs. Also, this method can be operated at moderate temperature. The chemical oxidative polymerization is a two-electron exchange process; hence it needs oxidant for oxidizing monomers. For oxidative polymerization, the chloroaurate acid (HAuCl$_4$) [35], ammonium peroxydisulfate (APS) [36], FeCl$_3$ solution [37], etc. have been extensively used as oxidizing agents.

In addition to the oxidative polymerization, the second most favourable synthesis method is electro-polymerization. In this process, the polymerization occurs on the working electrode via application of external voltage. The synthesis is usually carried out inside the glove box. The working electrodes used in electro-polymerization are either ITO-coated glass slides or gold-coated glass etc. [38]. The acetonitrile (MeCN) and propylene carbonate solvent are highly preferred solvents used in electro-polymerization methods. Furthermore, LiBF$_4$ (lithium tetrafluoroborate), TBABF$_4$ (tetra-n-butylammonium tetrafluoroborate), TBAClO$_4$ (tetra-n-butylammonium perchlorate), TBAPF$_6$ (tetra-n-butylammonium hexafluorophosphate), LiClO$_4$ (lithium perchlorate), etc. have been used as supporting electrolytes [39–42].

In this context, Chen *et.al.* [43] reported the synthesis of poly-PNBTH-Boc. (PNBTH-Boc: 3,8-di([2,2′-bithiophen]-5-yl)-6-oxophenanthridine-5(6H)-carboxylate) from monomer PNBTH-Boc via electro-polymerization using 0.9 V positive pulse potential for 30 s on working electrode. Figure 1 represents the schematics of synthesis of poly-PNBTH-Boc.

In addition to the above-mentioned π-CPs, the derivatives of π-CPs have high potential as electrode material for ESS. The derivatives of π-CPs have been highly employed as electrodes in ESS. The derivative of PANI such as POTO (Poly(o-methylaniline)), POAS (poly(o-methoxyaniline)) and PDMA (poly(2,5-dimethylaniline)) are highly implemented in versatile ESS. [44] Moreover, the PTh derivative PMeT (poly(3-methylthiophene) is also used as active electrode material in ESS [7,45].

The presence of conducting materials, multiwalled carbon nanotubes (MWCNTs) and single-walled carbon nanotubes (SWCNTs) in π-CPs influence the final oxidation ratio of the polymer chains and change the properties of substituent attached to aromatic rings. To reflect this fact, Bavastrello *et al.* have synthesized MWNTs and SWNTs inserted poly(o-methylaniline), poly(o-methoxyaniline), poly(2,5-dimethylaniline), poly(2,5-dimethoxyaniline) via oxidative polymerization and fabricated thin films using the Langmuir–Schaefer technique [44].

The interaction of carbon nanotubes (CNTs) with π-CPs increases the average localization length which further enhances the electrical conductivity of composite. To demonstrate this fact, using p-toluenesulfonic acid as surfactant dopant and APS as initiator, Imani *et al.* [46] have successfully synthesized PPy-MWCNTs nanocomposites via in situ oxidative polymerization of pyrrole. Furthermore, this study demonstrates the electrical conductivity of increases with increasing the MWCNTs contained in composite.

$H^+$ deficiency of PANI-based electrodes is responsible for the fading of their redox reversibility and results as a serious issue in terms of cycling instability. The addition of pH buffers in electrolytes is one of the ways to improve this issue. Liu *et al.,* using ammonium persulfate as oxidizing agent and oxidative polymerization method, have fabricated CNTs–PANI–PEDOT material for cathode for zinc-ion batteries. The use of multi π-CPs CNTs composites enhances the electrochemical reactivity and electrode stability. The presence of $SO_3^-H^+$ groups in PSS contributes to the improvement in electrochemical properties and reversibility properties of CNTs–PANI–PEDOT cathode. The observed improvement may be due to the additional protonation of PANI generated by the presence of $SO_3^-H^+$ group in the cathode framework [47].

The dissolution and the volume change of π-CPs during charging and discharging is one of the serious issues as cathode material for ESS. For improvements of unsatisfactory cycling stability and improved practical applications of π-CPs, recently, the fabrication of composite with like graphene is highly recommended. The graphene is highly porous and highly conductive in nature. Jin *et al*. have demonstrated the enhancements of charge mobilization in electrolyte and overcomes the dissolution of π-CPs via fabricating π-CPs and graphene composite. For this study, using the electrochemical polymerization method, they have synthesized the nitrogen-doped porous graphene film-supported vertically aligned PANI nanocones as a cathode material for flexible SCPs. The obtained porous morphology of cathode material enhances the strong π-π stacking interaction in PANI relieves volume changes of PANI during the charging–discharging process [48].

# 3. Applications of π-conjugated polymer-based materials

## 3.1. Battery

With increasing pollution problems associated with vehicles having internal combustion engines, and the advancements in wearable and portable electronic gazettes, the developments of electrochemical energy storage batteries are highly promoted. The applications such as electric vehicles (EVs), portable electronics, and for frequency regulation, etc. highly demand the batteries with low cost, high power density, light weight, high safety, long life and environmentally friendly in nature [12,49–53]. The electrochemical energy storage batteries include sodium sulfur (Na-S), sodium-ion (Na-ion), K-ion), Zn-ion, Zn-air, lithium-ion (Li-ion), lithium air (Li-air), redox flow, aqueous rechargeable and thin-film batteries. In this section, we have illustrated the recent progress and advancements related with the π-CPs as electrode material and their electrochemical performance. Over the inorganic compounds, the electroactive organic compound has various attractive properties, which include light weight and greater safety [54–57]. Moreover, the electroactive organic compound can be synthesized with desired structure and appropriate functional group, which further provides more redox active sites for greater electrolyte ion exchange during the charging/discharging process [58–60]. More interestingly, the heteroatom with lone pair of electrons (viz. O, N doping in π-CPs) enhances the redox activity in solid/ liquid electrolyte [61,62]. Also, the availability of π-conjugation can facilitate the charge transport at electrode/electrolyte interface and enhance the intermolecular interactions [63]. These

properties of electroactive organic compounds make them more viable for high-energy density energy storage application.

From the last four decades, π-CPs; PANI [64], PPy [65], PTh, PI, [66] and their derivatives have been highly investigated and are used as electrode material in most of the ESS. Recently, with ever increasing new alternatives for high-energy density devices, the other novel π-CPs such as derivatives of PEDOT [67,68], derivatives of polyacetylene (PAc) [16,65], poly (3-hexylthiophene) [69], poly (hexaaza trinaphthalene) (P3HT) [15] and poly{[N,N′-bis(2-octyldodecyl)-1,4,5,8-naphthalenedicarboximide-2,6-diyl]-alt-5,5′-(2,2′-bithiophene)} (P(NDI2OD-T2)) [70] have emerged as promising potential candidates as an active material for anode/cathode in different metal-ion batteries (MIBs) and energy storage applications.

### 3.1.1. Lithium-ion batteries

Currently, for EVs, portable and lightweight electronics (viz., mobile phones, laptops, wearable electronics, etc.) and high grid scale energy storage highly demands the utilization of the LIBs [71,72]. The LIBs dominate over all other commercialized rechargeable batteries. The electrode, binder, separators and electrolyte are the main components of the LIBs. Out of these components, the electrode is the key constituent which decides the output power and energy density of the LIBs. The copper/aluminium is used as a conducting substrate. The anode of LIBs is fabricated from the Li-based oxide as active electrode material, whereas still, the cathode of LIBs depends on lithium-containing inorganic materials. However, the volume expansion, pulverization and phase transition occurs in inorganic which decreases the cycle life and energy density of LIBs. Moreover, dendrite formation of inorganic material in LIBs is also a serious issue, which can lead to explosion. Hence, there is an urgent need for an alternate material for a positive electrode (cathode) material of Li-ion battery [73–75].

The organic polymer material is one of the important classes of material, which shows many attracting properties such as abundant availability, high redox activity, recyclability and environmentally benign [15,52,76–78]. The different polymer compounds, which includes conducting polymers [78,79], carbonyl compounds [80], organic free-radical compounds [81], organosulfur compounds: disulphite [18], imine compounds [82], azo compounds [83], etc. are extensively employed as active material for the cathode in the LIBs. The π-CPs show many attracting properties such as extended π-conjugated system, good electrical conductivity, porous structure, high ionic diffusion, low cost and environmentally friendly nature. Owing to such outstanding properties, the π-CPs are considered as novel materials and are widely employed as the cathode material for LIBs [47,58,59,65,70,84,85]. The different π-CPs such as PANI, PPy, PTh, PI and PEDOT have been extensively used as cathode material for LIBs. The reversible capacity of LIBs strongly depends on the dopant and doping level of $Li^+$ and/or anions in the π-CPs. By principle, at higher doping level, the π-CPs should provide the higher reversible capacity. However, at higher doping, side reaction and instability of the π-CPs limits its reversible capacity [58].

At high current density conditions, the LIBs limit its high-energy storage. To enhance the energy storage capacities of LIBs, Xie *et al.* proposed the electrode design containing hybrid materials as an active material for LIBs which combines the battery-like faradaic behaviour with the capacitor-like non-faradaic effect. For that they have synthesized two π-conjugated compounds, namely $Ni[C_6H_2(NH)_4]_n$:(Ni-NH) and $Ni[C_6H_2(NH)_2S_2]_n$:(Ni-S) as a battery material Ni-NH demonstrated initial capacity of 1655 mAh g$^{-1}$, reversible capacity of 1195 mAh g$^{-1}$ and excellent average Coulombic efficiency (CE) of 98.44% [84].

The development of the anode material with higher capacity and longer cycle life is one of the most important issues for the next generation of LIBs. In addition, the crumbling and cracking of the electrode leads to electrical disconnection from the substrate. Recently, considering the advantages of π-CPs as high conductivity, strong π–π interactions with nano-structured carbon and strong electrostatic interaction with transition metal agents, the composite of nano-structured carbon and nano-structured metal oxides with π-CPs has been extensively used as anode material for LIBs. For example, Hu *et al.* [86] studied the synergistic effect between the PMo, PANI and MWNTs. To explore the synergistic effect of PMo, PANI and MWNTs, they have fabricated the different composites of PMo, PANI and MWNTs as PMo12/PANI/MWNTs and PANI/MWNTs, and reported their discharge capacity of 1572 mAh g$^{-1}$ and 920 mAh g$^{-1}$, respectively. In addition, the obtained discharge capacity was found to be constant for 100 cycles.

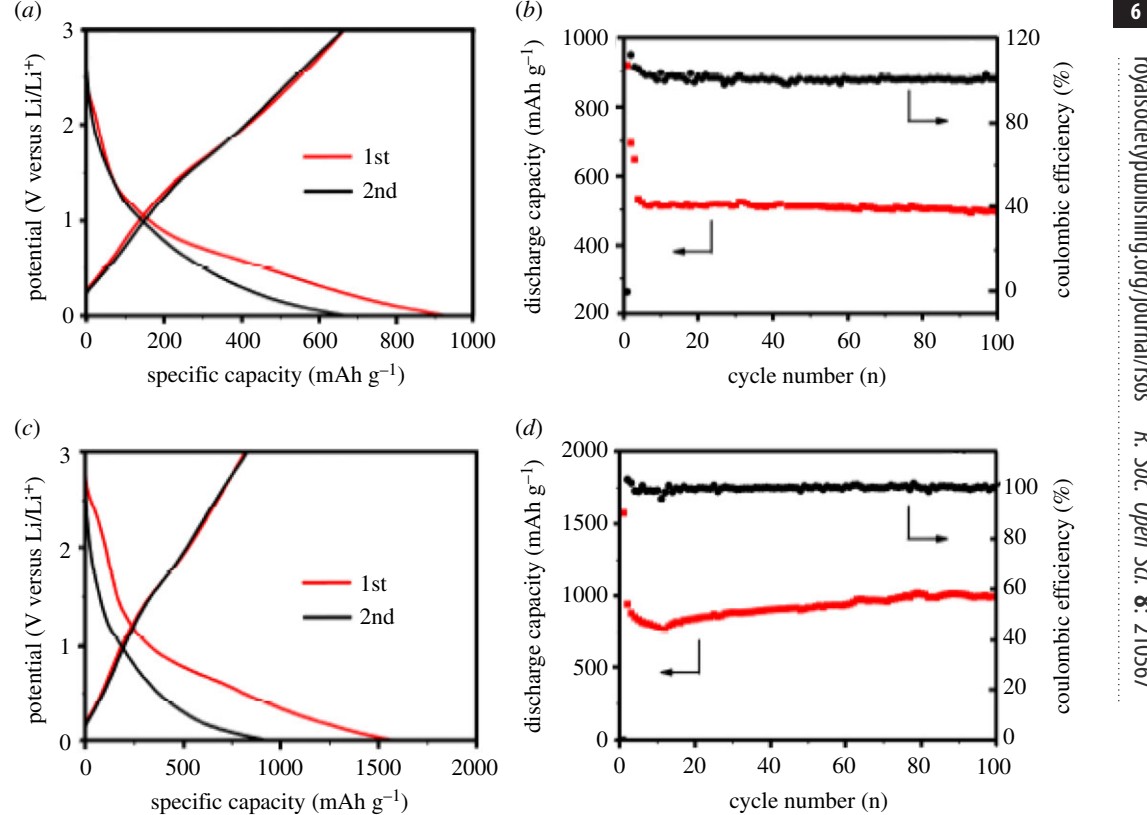

**Figure 2.** Charging and discharging curves of the lithium rechargeable battery for the first and second cycle $I = 0.5$ mA cm$^{-2}$: (a) PANI/MWNTs, (c) PMo12/PANI/MWNTs; discharge capacity and Coulombic efficiency versus cycle number; $I = 0.5$ mA cm$^{-2}$: (b) PANI/MWNTs and (d) PMo12/PANI/MWNTs [86]. Images reproduced with permission.

Figure 2a,c shows the charging–discharging profiles of the PMo12/PANI/MWNTs and PANI/ MWNTs, respectively, with the current density of 0.5 mA cm$^{-2}$ for the first two cycles. The measured first discharge capacities are 1572 mAh g$^{-1}$ and 920 mAh g$^{-1}$ for PMo12/PANI/MWNTs and PANI/ MWNTs electrodes, respectively, however the relevant discharge capacities are reduced to 942 and 693 mAh g$^{-1}$, respectively. For the PMo12/PANI/MWNTs nanocomposite, the reported value of the first discharge capacity is 1572 mAh g$^{-1}$, which decays up to 866 mAh g$^{-1}$ after 12 cycles. The discharge capacity increases again to 1000 mAh g$^{-1}$ and found to remain constant for the next 100 cycles. The value of capacity retention was found to be nearly 63.6% (figure 2b,d).

In particular, reflecting this fact various recent reports has shown the many advances in synthesis and adaptation of π-CPs as electrodes in LIBs. Although the many advances provided the π-CPs as cathode/ anode material for LIBs, the volume expansion, sloping of charge discharge profile, low cycleability etc. are still major limitations for π-CPs base LIBs. In addition, the summary of representatives for π-CPs used as electrodes in LIBs and their electrochemical characteristics are demonstrated in table 1.

### 3.1.2. Zn-ion batteries

Layered oxides, Prussian blue analogues, poly-anion and organic compounds are the four important materials used as electrode material for Zn-ion batteries (ZIBs) [91–93]. The main issues of concern with the use of above-mentioned electrode material in ZIBs are low specific energy, and low rate capability and less stability [94–96]. Therefore, there is an urgent need to find suitable candidates for ZIBs which can enhance the discharge capacity with rate capability in solid as well as liquid electrolytes [97]. Due to the remarkable redox activity and moderate electric conductivity of π-CPs, recently the use of different π-CPs (viz. PANI, PPy PEDOT, PSS, PTh, etc.), their derivatives and composite with CNTs as the cathode material for ZIBs have been considerably increased.

In this context, Liu et al. [47] have developed a CNTs–PANI–PEDOT:PSS and used it as cathode material for ZIBs. In this study, the use of multi π-CPs composite with CNTs introduced synergistic effect among π-CPs and enhanced the electrochemical performance in polyacrylamide hydrogel

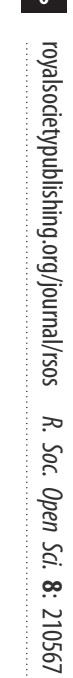

**Table 1.** Representative π-CPs and their performance in different MIBs.

| sr. no. | type of battery | material | electrolyte | potential window (V) | reversible capacity (mAh g$^{-1}$), current density (A g$^{-1}$) | capacity retention (cycle number, current density) (A g$^{-1}$) | reference |
|---------|-----------------|----------|-------------|----------------------|---------------------------------------------------------|----------------------------------------------------------|-----------|
| 1. | magnesium ion battery (MgIBs) | PI\CNT | Mg(HMDS)2-4MgCl2/2THF-PP14TFSI | 0.5–2.5 | 161 at 1C | 57% (8000) (10 C) | [87] |
| 2. | SIBs | PHATN | 4 m NaPF6/DME | 1–3.5 | 220 (50) | 83.8% (50 000) (10) | [15] |
| 3. | sodium-sulfur batteries (SSBs) | RGO-g-P3HT | 1 M LiC2F6NO4S2/LiNO3 | 1.5–3.0 | 1288 (0.05 C) | 59.4% (100) (0.2 C) | [69] |
| 4. | LIBs | P(NDI2OD-T2) | 1 M LiClO4 in DME | 0–2 | 54.2 | 95% (3000) (10 C) | [70] |
| 5. | LIBs | Ni[C6H2(NH)2S2]n (Ni-S), | Li | 3.0–0.005 | 1164 at (0.1) | 98.44% (1500) (3) | [84] |
| 6. | ZIBs | CNTs–PANI–PEDOT:PSS | PAM | 0.5–1.6 | 238 (0.2) | 100%. (10) | [47] |
| 7. | ZIBs | CNTs–PANI–PEDOT:PSS | 2 M ZnSO$_4$ | 0.5–1.6 | (238) at (0.2) | 100%, (1500), (1000) | [47] |
| 8. | hybrid-flow batteries (HFBs) | PAQPy/G | 0.1 M NaAc | 0.5–1.5 | 62.2 (0.2) | 74.5% (100) (1) | [88] |
| 9. | aqueous polymer-air battery (APBs) | P14AQ/CNT | 6 M KOH | −1 to −0.4 | 147 mAh (38.5 C) | 92% 500 cycles with | [89] |
| 10. | SIBs | PYT-TABQ/rGO | 1 M NaPF6 in DME | 1.0–3.5 V. | 210, (0.2) | 98% (1) (1400) | [90] |

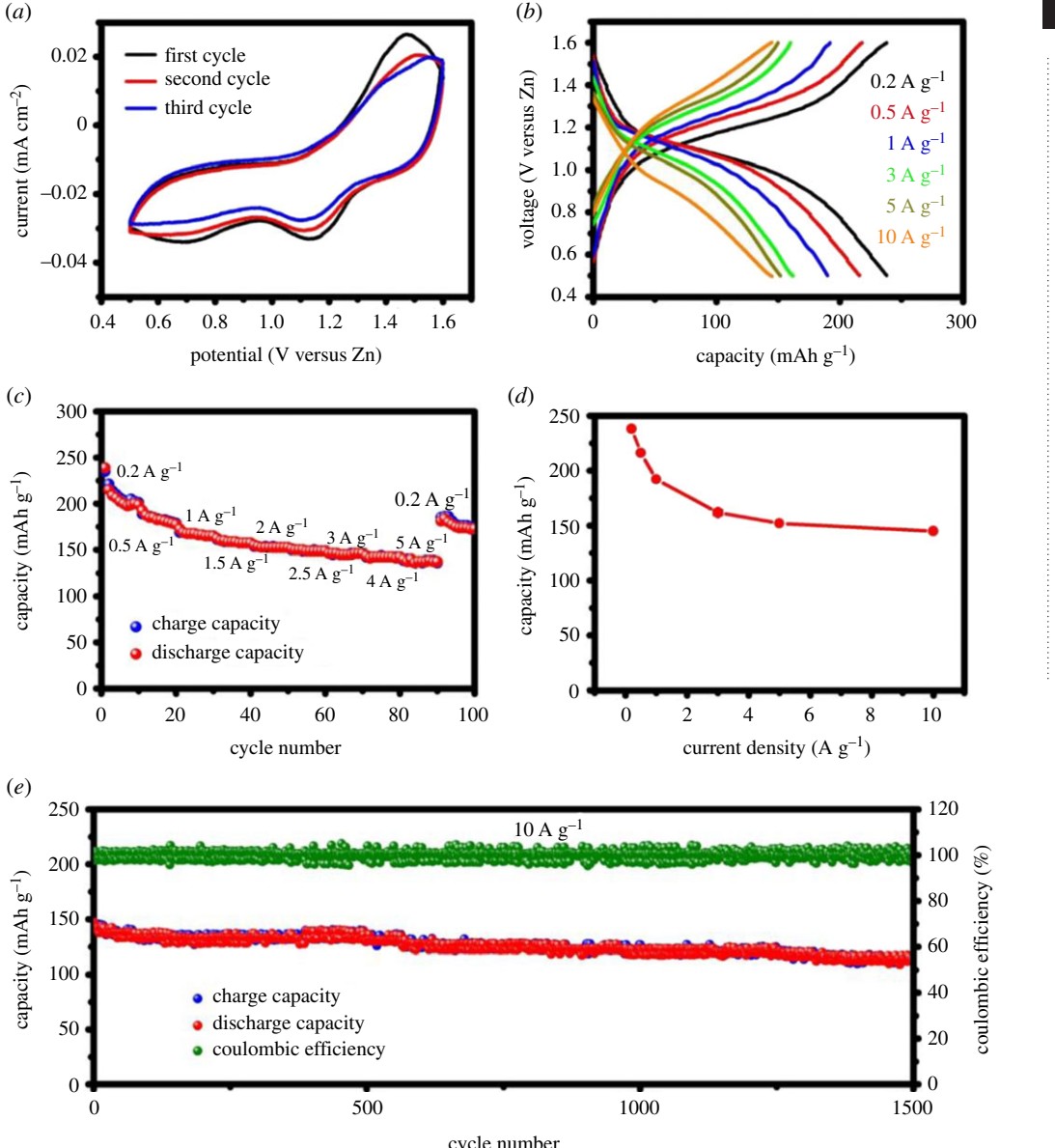

**Figure 3.** Electrochemical performance of the t-CNTs–PA–PE cathode in ZIBs. (*a*) CV curves; (*b*) GCD curves; (*c*) and (*d*) rate performance; (*e*) cyclic performance at 10 A g$^{-1}$ [47]. Images reproduced with permission.

electrolyte, and further reported the discharge capacity of 238 mAh g$^{-1}$ at a current density 0.2 A g$^{-1}$ with high-rate capability over a large number of cycles. In this study, the electrochemical activity and reversibility of ZIBs improved due to insertion of $-SO_3^-H^+$ groups in PSS, which provides large numbers of H$^+$ for the protonation of PANI in cathode. Also, the use of CNTs with π-CPs enhances the electrical conductivity of the CNTs–PANI–PEDOT:PSS electrode in ZIBs. The GCD profiles of the CNTs–PANI–PEDOT:PSS electrode under various current densities shows the charge/discharge plateaus (figure 3*b–d*). In addition, the battery demonstrated an excellent rate capacity because of the high conductivity of the t-CNTs–PA–PE cathode. The achieved capacity was found to be 145 mAh g$^{-1}$ even at a high current density of 10 A g$^{-1}$ (figure 3*d*).

### 3.1.3. Lithium-sulfur batteries

Alternative to the LIBs, the lithium-sulfur batteries are gaining considerable attention because they deliver energy density which is found five times more than the existing LIBs [98,99]. For the fabrication of the cathode, various organometallic materials are investigated. Out of different polymer

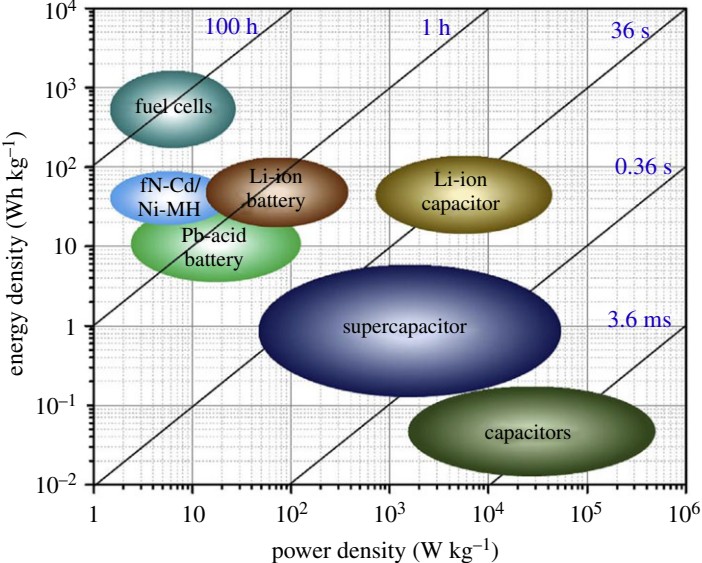

**Figure 4.** Ragone plot of different energy storage devices [1]. Images reproduced with permission.

materials, the π-CPs is one of the important classes. For the high flexibility and light weight, the π-CPs are highly preferred [76,100,101].

### 3.1.4. Potassium-ion batteries

Considering the abundant and widespread availability, high electrochemical conductivity, larger transfer number, higher mobility of potassium K-ion, over the last decades, there is tremendous increase in the research interest in K-ion as cathode material for potassium-ion batteries (KIBs). Due to the excellent performance of K-ion batteries with different anode materials like graphite, SWCNTs, MWCNTs and conducting polymers. Today, the K-ion batteries have emerged as complementary technology to Li-ion batteries and offer a significant research opportunity as a versatile energy storage device. The K-ion battery demonstrated excellent kinetics, high-rate and low-cost energy storage application. The research on K-ion batteries is in its early stage. Therefore, the major objectives among the researchers are the development of suitable cathode material and electrolyte for K-ion batteries demonstrated high-energy density of K-ion batteries than LIBs and sodium-ion batteries (NIBs) [1]. The large ionic radius of K-ion, low melting point, large volume expansion, structural deterioration and extremely high activity of K-ion limit has applicability as electrode (positive) material for KIBs.

### 3.1.5. Sodium-ion (Na-ion) batteries

Owing to the abundance, low cost as well as environmental benign nature of sodium (Na) sources compared with Li metal, the NIBs demonstrate great potential as a promising alternative to the LIBs. In the last two decades, the Nickel (Ni) and Cobalt (Co) are studied extensively as the cathode material for NIBs. However, these metals are also high cost and result in a very low reversible capacity [15]. Hence, there is a need for an electrochemically active, more economic, light weight, environmentally benign electrode material for NIBs. In NIBs, different organic materials such as organic free-radical compounds, carbonyl compounds, organosulfur compounds, imine compounds [102] and azo compounds [103] have been used as electrodes for NIBs and other batteries. Out of the different organic materials, recently the π-CPs have gained enormous attention as cathode material for NIBs. Moreover, for a more detailed discussion of π-CPs materials and their application in NIBs, the reader is referred to recently published reviews [24,63,101,103]. In addition, the recent advances and highlights of representative π-CPs as an electrode material for NIBs are demonstrated in table 1.

## 3.2. Applications of π-conjugated polymers for supercapacitors

SCPs, also known as ultracapacitors, demonstrate the high power and moderate energy density for most of the portable consumer electronic devices (viz. laptop, wearable electronics, mobile phone, watches, roll display, etc.). In addition, the SCPs are also used in large backups and most awaited electrical vehicles.

[104–107]. Figure 4 demonstrated the Ragone plot of different ESS, which reveals the status of different ESS as a function of power density versus energy density. This shows the SCPs have greater energy density than conventional capacitors and lower than fuel cells and LIBs.

Based on the electrode material used in SCPs, it can be categorized into electrochemical double capacitors (EDLCs), pseudocapacitors and hybrid SCPs [60,61]. The electrochemical EDLCs employ carbon-based material as an electrode. The pseudocapacitors use transition metal-based oxides, hydroxides and sulfide, and polymers as an electrode in it, whereas the hybrid SCP is the superposition of EDLCs and pseudocapacitor [11,104,108,109].

### 3.2.1. Pseudocapacitors

In many advanced applications, the low-energy density of the SCPs limits its practical applicability as an energy storage device [110–112]. In fact, the energy density ($E = 1/2\,CV^2$) of the SCPs is linearly dependent on the specific capacitance as well as the square of the operating potential window used for study [43,47,77] To date, researchers have adopted many advanced strategies to increase the specific capacitance of the SCPs, which includes strategies to extend the operating potential window of SCPs and to fabricate the electrode using various high surface area active materials, such as carbon, carbon SWCNTs, MWCNTs, graphene, carbon aerogel, metal oxides hierarchical nano-structured, hierarchical meso- and micro-porous microstructures, organic electroactive molecules as electrode for SCPs [14,88,109,112–115]. To increase the operating voltage of SCPs, the new material as electrode in SCPs which can operate at high voltage and deliver more power is a prime requirement. In addition, the use of high voltage electrolyte which can sustain high voltage has an equal importance. In this regard, over the past three decades, the use of π-CPs as the electrode material for SCPs has attracted extensive interest due to their attractive properties such as low cost, less volume expansion and less structural variation in organic/polymer electrolyte, abundant availability, structural diversity, environmentally benign, etc. [13,17,78,116,117]. More interestingly, the structural diversity and facile molecular design of π-CPs can modify the electrochemical performance of SCPs [58]. Therefore, considering the above-mentioned merits of the π-CPs, they have been extensively studied as electrodes in SCPs. The various π-CPs such as PPy [118], PTh [43], PANI [119], poly(p-phenylene) (PPhP) [120] and PAc and PEDOT [6] have been extensively explored as active electrode materials for SCPs.

In this regard, Wang et al. [121] studied a new Faradaic electrode donor/acceptor-CP material poly(3,4-ethylenedioxythiophene):poly(styrenesulfonate) PEDOT:PSS, (PSS: poly(styrenesulfonate)) as the cathode and poly(4-(4,4-dihexadecyl-4H-cyclopenta [2,1-b:3,4-b′] dithiophen-2-yl)-6,7-di(thiophen-2-yl)-[1,2,5] thiadiazolo[3,4-g] quinoxaline) (PCQTh) as the anode for SCPs. The narrow bandgap existing in PEDOT:PSS enhances an intrinsic electrical conductivity and charge delocalization in the reduced state. In this study, the intrinsic electrical conductivity at charge neutral state for PCQTh is reported to be approximately equal to $10^{-3}\,S\,cm^{-1}$ in the absence of dopants. The two redox peaks in the negative potential range in the cyclic voltammetry curve of the PCQTh electrode demonstrated that there are two-electron-transfer processes between 0 and −2 V. The molecular orbital calculations of spin densities projected the high degree of charge delocalization throughout the polymer chain; this reveals that there were no active redox sites while oxidizing and reducing the polymer; however, some of the positive charge density increases at donor unit and some of the negative charge density increases at acceptor unit.

Low electrical conductivity and low electrochemical activity of different π-CPs limits its practical applicability in various electrical applications. For example, PEDOT is one of the important π-CPs with high electrical conductivity of the order of $1000\,S\,cm^{-1}$. Nevertheless, the PEDOT shows relatively low electrochemical activity in comparison to PANI and PPy. Therefore, to combine high electrochemical activity with high electrical conductivity, the formation of composites of high conductive π-CPs (e.g. PEDOT) with high electrochemical active π-CPs (e.g. PANI/PPy) is highly preferred. For example, Yang et. al. fabricated PEDOT/PANI (figure 5), and studied the PEDOT/PANI π-CPs hydrogels as a freestanding electrode for flexible solid-state SCPs. For the fabrication of PEDOT/PANI, the phytic acid used as a molecular bridge enhances the π–π interactions between PEDOT and PANI chains and permits the incorporation of PEDOT and PANI at a molecular level. The PEDOT/PANI electrode demonstrates the excellent electrochemical behaviour in PVA/H$_2$SO$_4$ gel electrolyte and resulted in the highest specific capacitance of $112.6\,F\,g^{-1}$ at a scan rate at $5\,mV\,s^{-1}$ [13].

Figure 6a shows the schematic of PEDOT sheets containing PANI and shows different magnification SEM images of PEDOT/PANI hydrogel. During the charge–discharge process the swelling and shrinking effect is observed in PANI, which further decreases the electrical conductivity of PANI. Whereas, in this study, PEDOT/PANI hydrogel is reported to be electrochemically stable and showed

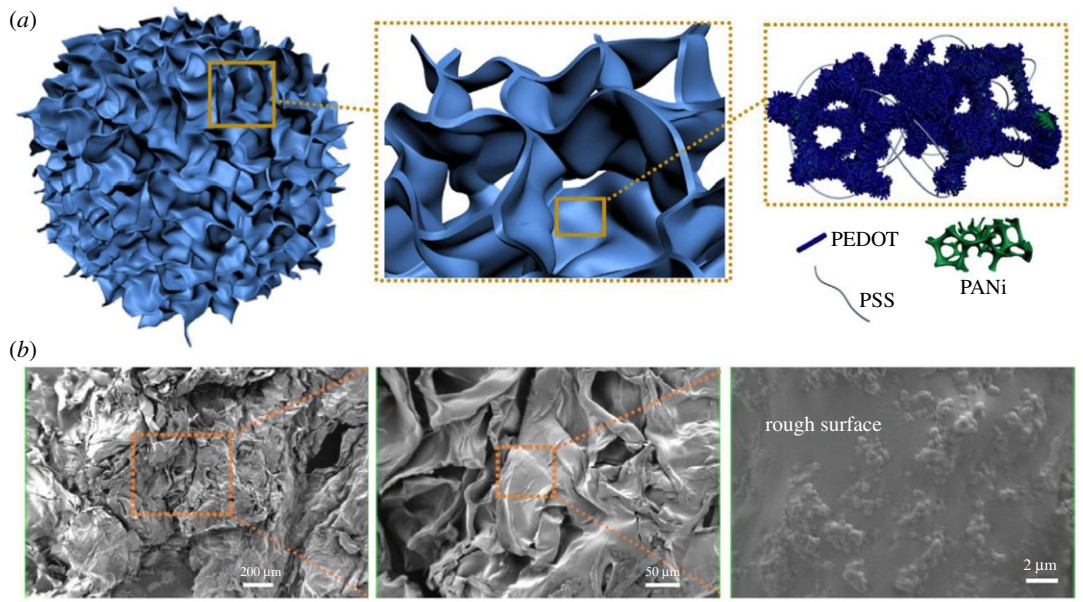

**Figure 5.** (*a*) Schematic of hydrogel skeleton, porous network and PEDOT sheets containing PANI. (*b*) SEM micrographs of lyophilized PEDOT/PANI hydrogel at different magnification. The rough surface of the sheet evidences the inlay of PANI particles in each PEDOT sheet [13]. Images reproduced with permission.

the capacitance retention of 80.8% after 5000 galvanostatic charge–discharge cycles. Interestingly, PEDOT/PANI hydrogel SCPs demonstrate a volumetric energy density of 0.25 mWh cm$^{-3}$ at a power density of 107.14 mW cm$^{-3}$. In addition, the summary of representative π-CPs and electrochemical performance as an electrode for SCPs is demonstrated in table 2.

### 3.2.2. Hybrid capacitors

To enhance the specific capacitance and energy density of SCPs, various research efforts have been focused on exploring the novel electrode and electrolyte materials, adoption of various synthesis strategies to obtain materials with high surface area, uniform porosity, and enhancement of the potential window, etc. Use of the π-CPs with carbon-based material (for example: graphene quantum dots, graphene, CNT, etc.) as a hybrid electrode in SCPs emerged as a promising approach for high-energy storage applications [104]. Among the various polymers, π-CPs are the most attractive material and can form either covalent or non-covalent composite with carbon-based materials which produces the synergistic effect and results in high-energy density. In addition, due to the Faradaic as well as non-Faradaic charge storage capability of conducting π-CPs results in higher specific capacitance compared with EDLCs [77,126,127]. In addition, the π-CPs shows strong π–π interactions with carbon material [86]. The synergetic effect and chain-ordering are the main factors in hybrid electrodes which are responsible for high electric conductivity and charge at the electrode electrolyte interface. The in hybrid electrode occurs due to the chain expansion by the chemical interactions between π-CPs and the solvent induces the synergistic effect, whereas the chain-ordering occurs due to the π–π conjugation between π-CPs and carbon-based material [128,129]. Recently π-CPs have been used in hybrid capacitors.

For example, Kaushal *et al*. [130] have fabricated the PANI/CNTs/graphene, MnO-doped PANI/CNTs and MnO-doped PANI composites via in situ redox deposition. The as-prepared composites further evaluated for their electrochemical performance for hybrid SCPs. Out of the above-mentioned composites, the PANI/CNTs/graphene composite showed the higher electrochemical performance and resulted in high specific capacitance of 1360 F g$^{-1}$ at 5 mV s$^{-1}$ scan rate, and cyclic stability of 85% over 5000 Galvanostatic charge–discharge cycles. Interestingly, this obtained specific capacitance is 30% higher than MnO$_2$-doped PANI/CNTs and 50% higher than of MnO$_2$-doped PANI composite. In addition, the representative summary of π-CPs and their electrochemical performance for hybrid SCPs demonstrated in table 3.

### 3.2.3. Asymmetric capacitors

Low-energy density of SCPs limits its applicability in various recent advanced energy storage applications such as hybrid EVs, portable electronics and power backups. Out of the different

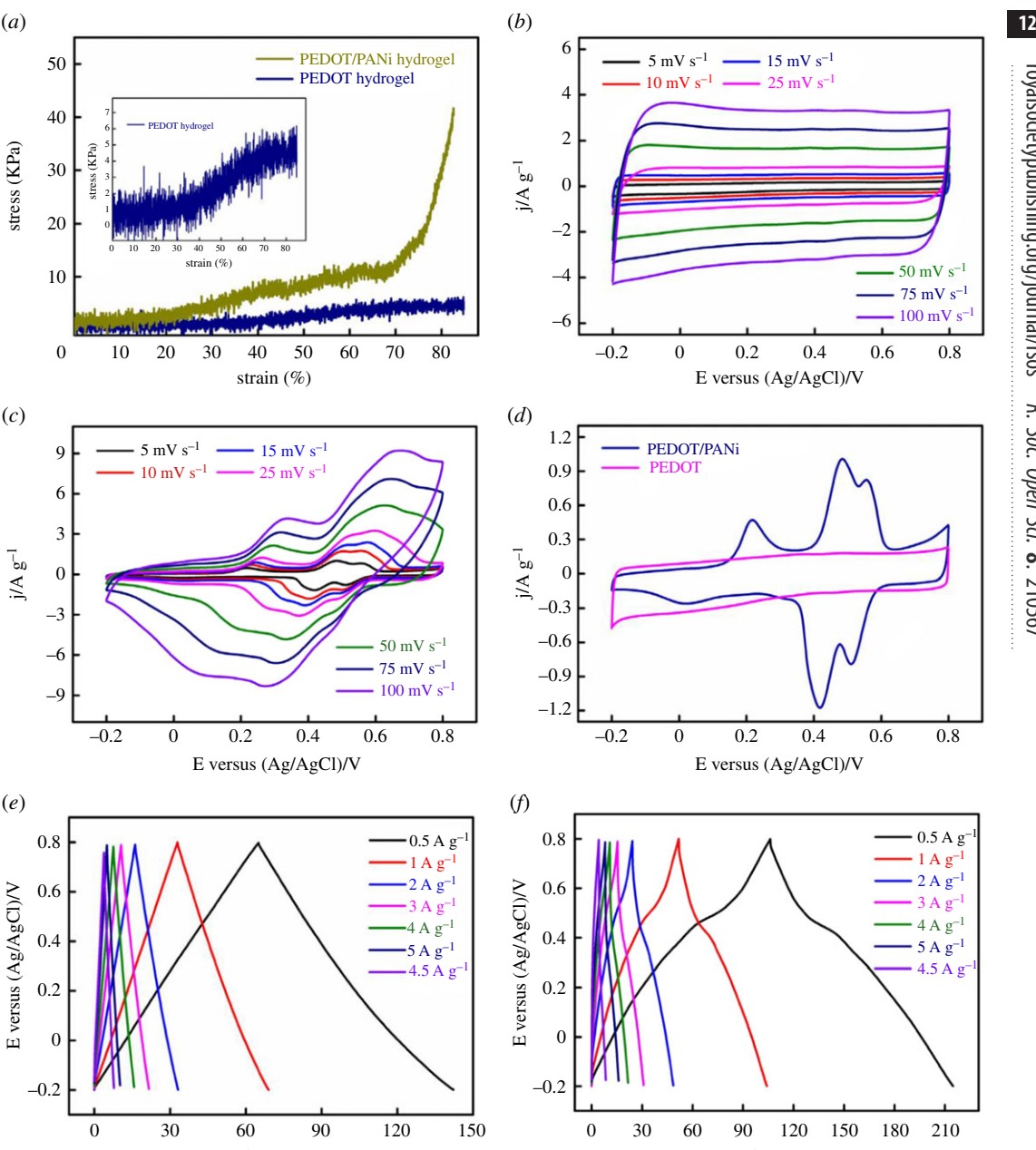

**Figure 6.** (*a*) Compressive stress–strain curves of PEDOT hydrogel and PEDOT/PANI hydrogel. CV curves of (*b*) PEDOT hydrogel and (*c*) PEDOT/PANI hydrogel at different scan rates. (*d*) CV curves of PEDOT and PEDOT/PANI at 5 mV s$^{-1}$. GCD curves of (*e*) PEDOT hydrogel and (*f*) PEDOT/PANI hydrogel at different current densities [13]. Images reproduced with permission.

approaches used to increase the energy density of the SCPs, increasing the working potential window via fabricating asymmetric SCPs with dissimilar electrodes are the most promising approaches used. For the fabrication of asymmetric SCPs, the anode and cathode material are fabricated independently. Moreover, for electrochemical demonstration, each electrode in asymmetric SCPs can operate at different operating voltage windows. Furthermore, this operation of electrodes at different potential windows enhances the overall operating potential windows of asymmetric SCPs, which ultimately enhances the energy density of asymmetric SCPs. [3,48,60]. The electrochemical performance of asymmetric SCPs is found to be higher than carbon-based SCPs [14,43,121,137]. π-CPs are the potential candidates for asymmetric SCPs. Up to date, many studies have been reported pertaining to successful utilization of π-CPs as anode/cathode with other electrodes such as metal oxides, reduced graphene oxides and CNTs in asymmetric SCPs [43,59,117,121,130].

**Table 2.** Representative π-CPs and their performance in SCPs.

| sr.no. | material | method of synthesis | electrolyte | potential window (V) | specific capacitance F g$^{-1}$ at current density A g$^{-1}$/ scan rate | energy density Wh kg$^{-1}$ | capacity retention (cycle number), current density) (A g$^{-1}$) | reference |
|---|---|---|---|---|---|---|---|---|
| 1. | PCMPs | chichibabin reaction method | — | 0–1.0 | 324 (0.1) | — | >97% (10 000) (2) | [116] |
| 2. | G-PTEPE-TBPE | electrospinning | 6 M KOH | −1–0 | 179 (0.2) | — | — | [113] |
| 3. | PANI@Co-Porphyrins | | 1.0 M H$_2$SO$_4$ | 0–0.6 | 823 (0.5) | 41 | 91% (1000) (5) | [17] |
| 4. | BVO4/PANI | in situ chemical oxidative polymerization | 1 M KOH | 0–0.66 | 701 (1) | — | 95.4% 5000 1 | [122] |
| 5. | co-MOF/PANI | in situ chemical oxidative polymerization | 1 M KOH | 0 to 0.62 V | 504 (1) | — | 90% 5000 2 | [123] |
| 6. | PEDOT/PANI | molecularly bridging via phytic acid | (PVA)/H$_2$SO$_4$, gel-electrolytes | 0.2–0.8 | 112.6, (5) | 0.25 | 80.8% capacitance retention after 5000 GCD cycles at 7.5 A g$^{-1}$. | [13] |
| 7. | PDT/Ti3C2Tx | | 0.5 M H$_2$SO$_4$ | 0.0–0.6 | 284 mF cm$^{-2}$, 50 mA cm$^{-2}$ | 24 | 100% after 10 000 | [110] |
| 8. | MWCNT-DAP-TCA-P(EDOT-co-PyMP) | CNT-TEPA@PEPy | 0.5 M, LiClO$_4$ | −0.5 to +0.5 V | (139.6) (0.5) | — | 72.5%, 20 000 | [124] |
| 9. | QOP-BOP | | M Na$_2$SO$_4$ electrolyte | −1.2–0.0 | 305 (2) | — | 88% (1000) | [5] |
| 10. | PEDOT | | 1.0 M H$_2$SO$_4$ | 0–0.1 | 90 (1) | 1.8 mWh cm$^{-3}$ | 93%, 15 000 cycles at 30 mA cm$^{-2}$ | [125] |

**Table 3.** Representative π-CPs and their performance hybrid SCPs.

| sr. no. | material | method of synthesis | electrolyte | potential window (V) | specific capacitance at current density/scan rate | energy density Wh kg$^{-1}$ | capacity retention (cycle number), current density) (A g$^{-1}$) | reference |
|---|---|---|---|---|---|---|---|---|
| 1. | PPy/Fe$_2$O$_3$/r-GO | hydrothermal synthesis followed by oxidative polymerization | (1 M) Na$_2$SO$_4$ | −1.0–0 | 140 (0.20 A g$^{-1}$) | 19.5 | 93% (5000) (1) | [85] |
| 2. | PANI-MWCNT-Ni(OH)$_2$ | chemical precipitation method | 1 M KOH | 0.0–0.49 | 1013 (1 A g$^{-1}$) | 33 | 75% (1000) (25) | [78] |
| 3. | NDC/PANI | in situ polymerization | 1 M H$_2$SO$_4$ needs | −0.2–0.8 | 686 (1 A g$^{-1}$) | | 87.8% (5000) (10) | [131] |
| 4. | CeO$_2$/PANI | — | 1 M HCl | 0 to +0.8 V | 1452 (2 Ag$^{-1}$) | 75 | ~0% (20) (1500) | [132] |
| 5. | NPG@PANI | electrochemical polymerization | 1 M pACM/Et4NBF4-AN | 1.9 to 0.55 | 330.2 (1 mA cm$^{-2}$) | 1.47 | 88.7% (10 000) (4) | [48] |
| 6. | PPY-GQDs | chemical polymerization | 1 M NaCl | −0.7–0.7 | 647.54 (—) | 93 | 91.7%, (1400) (0.17) | [133] |
| 7. | PYT-TABO/rGO | — | 1 M Na$_2$SO$_4$ | −0.8 to 0.0 V | 312.5 (1) | 73.3 µWh cm$^{-2}$ | 94.5% (10) (10 000) | [90] |
| 8. | PPy/RGO/CNT/BC | | (1 M) NaNO$_3$ | −0.3 to +0,5 V | (715 mF cm$^{-1}$) (1 mA cm$^{-2}$) | 0.0328 | 86.85% 5000 | [134] |
| 9. | PPy/MWCNT | | H$_2$SO$_4$ (0.5 M) 0 | 0 to +0.8 V | 292 (0.2 A g$^{-1}$) | 292 | 89.2% (1000 ) (1.0) | [135] |
| 10. | LiFe5O8@PPy | | 1 M LiNO$_3$ | −1 to −0.2 V | 292 (5 mV s$^{-1}$), | | 82.99% (10 000) | [136] |

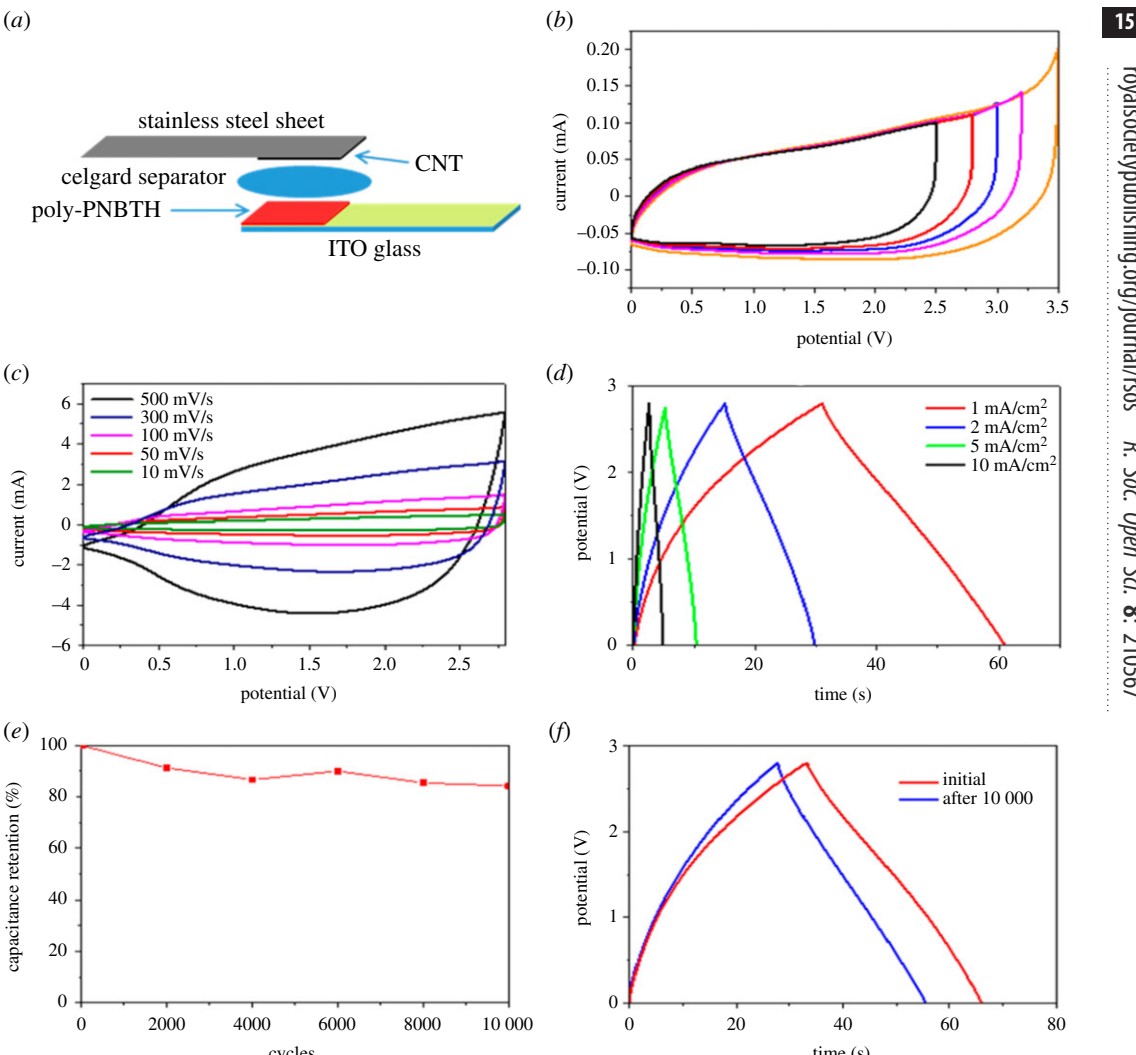

**Figure 7.** (a) Scheme of the asymmetric SCPs device (electrolyte: 1 : 1 volume ratio 1-buty l-1-methylpyrrolidinium bis-(trifluoromethanesulfonyl)imide:0.1 M tetrabutylammonium hexafluorophorophosphate in propylene carbonate). (b) CV spectra of the asymmetric SCPs under different electrochemical windows. (c) CV spectra of the asymmetric SCPs under different scan rates (voltage window: 2.8 V). (d) Galvanostatic charge/discharge results of the asymmetric SCPs under different current densities. (e) Cycling stability under galvanostatic charge/discharge conditions. (f) Comparison of galvanostatic charge/discharge profiles before and after 10 000 cycles [43]. Images reproduced with permission.

In this context, Zang *et al.* [138] fabricated asymmetric SCPs by adopting the ruthenium oxide-modified-reduced graphene oxide sheets as an anode and ruthenium oxide PANI as cathode (rGO–$RuO_2$//rGO–PANI). In comparison with the two symmetric SCPs fabricated using two rGO–$RuO_2$ or two rGO–PANI, the asymmetric SCPs fabricated using rGO–$RuO_2$ and rGO–PANI showed the considerable enhancement in the specific capacitance of 357 F g$^{-1}$ at current density 0.3 A g$^{-1}$ with high energy 12.4 Wh kg$^{-1}$. In addition, the asymmetric SCPs show excellent capacitive retention of about 70% after 2500 cycles. The schematics of the fabrication of the asymmetric SCPs device are demonstrated in figure 7a. To explore the effect of independent tailored anode and cathode and their synergistic effect in asymmetric SCPs, the cyclic voltammetry curves at different potential windows are demonstrated in figure 7b. Further, the Galvanostatic charge–discharge profile at different current densities and cyclic voltammetry curves at different scan rates of asymmetric SCPs are shown in figure 7c and d, respectively. In addition, the cyclic performance is demonstrated in figure 7e and f. Moreover, the summary of representative recent reports based on π-CPs used as an electrode material in asymmetric SCPs is demonstrated in table 4.

**Table 4.** Representative π-CPs and their performance asymmetric SCPs.

| sr.no. | material (mass ratio) | method of synthesis | electrolyte | voltage window (V) | specific capacitance at current density/scan rate | energy density Wh kg$^{-1}$ | retention of capacitance at (current density) (cycle numbers) | reference with year |
|---|---|---|---|---|---|---|---|---|
| | ACFC//PANI/CFC | | 1 M H$_2$SO$_4$ | 0–1.6 V | 808 F g$^{-1}$ | 36.35 Wh kg$^{-1}$ | 76% (1000) (5). | [139] |
| 1. | PCQTh //(PEDOT:PSS) | microwave-assisted cross-coupling copolymerization | 0.5 m tetraethylammonium tetrafluoroborate (TEABF4) in propylene carbonate (PC). | −0.2–1 | 103 (0.25) | 30.4 at 1 A g$^{-1}$ | 90% (2000) | [121] |
| 2. | (poly-PNBTH-Boc// CNTs | | PC/PYR14TFSI | 0–1.5 | 112.5 mF cm$^{-2}$ at 1 mA cm$^{-2}$. | 23.5 μWh cm$^{-2}$ at 1 mA cm$^{-2}$ | 85% (10 000). | [43] |
| 3. | PANi/PEDOT/PANi/ UrGO/PEDOT-MoS$_2$ | | PVA/H$_2$SO$_4$, g | 0.2 to −0.4 | 61 F cm$^{-3}$ 1 A cm$^{-3}$ | 5.4 mWh cm$^{-3}$, 1 A cm$^{-3}$ | 73% (1000) | [128] |
| 4. | GO@PANi/CNT-SS | | 1 M KCl | (1.0 to 1.6 V | 331.49 F g$^{-1}$ 5 mV s$^{-1}$ | 21.45 Wh kg$^{-1}$ | 83% (5000) | [140] |
| 5. | PEDOT/PBOTT-BTD | | 0.1 m TBAPF6/ACN solution containing 0.02 m EDOT. | | (21) (0.1 mA cm$^{-2}$) | 5.79 Wh kg$^{-1}$ | 86% (2000) | [141] |
| 6. | CuCo2O4@PPy//AC | — | 0.2 M KOH | 0.0–0.6 V | 2272 F g$^{-1}$ at 2 A g$^{-1}$. | 52 Wh kg$^{-1}$, | 96%, (20) (5000) | [142] |
| 7. | MoS2/MoO3/PPy// PPyNT/NDG | | 2 M H$_2$SO$_4$ | 0–1.5 | 84 F g$^{-1}$ at 5 mV s$^{-1}$. | 43.2 Wh kg$^{-1}$ | 126% after (5000) | [111] |
| 8. | AC/MWCNTs/MnO2/PPy | | 1 M Na$_2$SO$_4$, 0.2 M KI | 0.4–0.6 V | 806 (1 A g$^{-1}$) | 92.4 Wh kg$^{-1}$ | — | [132] |
| 9. | AC/NiCo2O4/NF@PPy | | 2 M KOH | 0.0–0.5 V | 1717 (g$^{-1}$ 2 A g$^{-1}$. | 68.9% Wh kg$^{-1}$ | 89.2% (10 000) (30) | [143] |
| 10. | Ppy@NiCo2S4/N-C | | 2 M KOH | 0–0.5 | 908.1 F g$^{-1}$ 1 A g$^{-1}$), | 50.82 Wh kg$^{-1}$ | 126.6% (2000) | [144] |
| 11. | PVA/PEDOT//rGO, | | 1 A g$^{-1}$ in 1 M KOH | (1.0 to 1.6 V | 113.48 F g$^{-1}$ at 5 mV s$^{-1}$ | 21.45 Wh kg$^{-1}$ | 83% (5000) | [140] |

# 4. Outlooks and prospective

The π-CPs, their derivatives and composites with nano-structured metal oxides and carbon-based materials have been widely used as electrode material in versatile ESS. Owing to the advantage of the flexibility, low cost, environmental friendliness, structural diversity and ease of functionalization through nano-structured engineering π-CPs are highly appreciated material and are the potential candidates for SCPs and MIBs. However, the low capacity, sloping plateau, poor stability etc. are the major limitations behind π-CPs as electrode material in storage devices. In addition, the dissolution of the π-CPs electrode in aprotic electrolytes results in the quick capacity fading during discharging of the batteries and SCPs. To date, many strategies have been reported in literature to overcome such challenges. These include fabricating a salt of π-CPs with organic carbonyl compounds, forming covalent compounds of π-CPs with conductive materials and forming composite with nano-structured carbon/ metal oxides via non-covalent approaches [145]. In addition, the recently reported representative π-CPs and their electrochemical performance as an electrode in MIBs, SCPs and hybrid capacitors and asymmetric capacitors have been summarized in this review. Likewise, the low intrinsic electrical conductivity ($\sigma \approx 10^{-3}\,\mathrm{S\,cm^{-1}}$) of π-CPs are mostly responsible for the low power density of ESS. To improve the electrical conductivity of the π-CPs, it needs to fabricate composites with highly conductive nano-structured metal oxides and carbon-based materials. Since the lone pair also has high electron mobility, it has been found that π-CPs encompass the lone pair containing heteroatoms (viz. oxygen, nitrogen, sulfur, etc.), show high electrical conductivity and high redox activity, which further improve the energy density and power density of ESS. Hence, each π-CPs has its inherent rewards and deficiencies as an electrode material for MIBs and SCPs. However, the high safety, low cost and environmentally friendly nature of π-CPs are the important properties, which can make π-CPs a potential and competent candidate for fabrication of flexible and lightweight electrodes for future low cost, environmentally friendly and high-energy density ESSs [58].

Data accessibility. This article has no additional data.

Authors' contributions. S.J.U. and A.K. collated and drafted the manuscript. A.K. and S.P.M. formulated the content and structure of the manuscript. Y.K1 assisted in sorting out data obtained from the literature. M.G. and Y.K2 coordinated the study. All authors finally approved the publication.

Competing interests. The authors declare no competing interests.

Funding. The authors acknowledge the financial support received from the Science and Engineering Research Board (SERB), Department of Science and Technology, Government of India (sanction no. ECR/2016/001871) under the scheme Early Career Research Award. A.K. would like to thank Council of Scientific and Industrial Research (CSIR), India for financial support under SRA Pool Scientist scheme (Award no.: 13(9131-A)/2020-Pool).

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
