## [Peer Review File · Royal Society Open Science]

Review History

RSOS-210567.R0 (Original submission)

Review form: Reviewer 1

Is the manuscript scientifically sound in its present form?

No

Are the interpretations and conclusions justified by the results?

No

Is the language acceptable?

Yes

Do you have any ethical concerns with this paper?

No

Have you any concerns about statistical analyses in this paper?

No

Recommendation?

Major revision is needed (please make suggestions in comments)

Comments to the Author(s)

In this study authors explained about π -conjugated polymer and their use in metal ion batteries and supercapacitors. Authors provided the basic and known information and I am sorry to say that I can't accept this paper for publication in its present form. I am recommending authors to revise their manuscript and also do needed modification.

Table format is not good for publication and that need to modify with acceptable format.

I did not see any figures in the manuscript, of course it is reproduced with permission but for reader's it is difficult to understand the content without figures.

Review form: Reviewer 2

Is the manuscript scientifically sound in its present form?

Yes

Are the interpretations and conclusions justified by the results?

Yes

Is the language acceptable?

No

Do you have any ethical concerns with this paper?

No

Have you any concerns about statistical analyses in this paper?

No

Recommendation?

Major revision is needed (please make suggestions in comments)

Comments to the Author(s)

The manuscript provides an interesting update of pi conjugated polymers for energy storage applications. However, before publication, the following points need correction/clarification:

1. There are several grammatical errors throughout the manuscript. Those need to be corrected before the manuscript can be published.
2. Section 3.1.4 on KIBs is very unclear. A more specific elaboration of how the pi-conjugated polymers have been tried (and failed) will be beneficial for the reader.
3. The authors point out that pi-conjugated polymers have gained "enormous attention" in NIB systems as cathodes. It would be great if they could point out some notable performances in section 3.1.5.

Decision letter (RSOS-210567.R0)

Dear Dr Kumar:

Title: A review of π -conjugated polymer based nanocomposites for metal-ion batteries and supercapacitors

Manuscript ID: RSOS-210567

The editor assigned to your manuscript has now received comments from reviewers. We would like you to revise your paper in accordance with the referee and Subject Editor suggestions which can be found below (not including confidential reports to the Editor). Please note this decision does not guarantee eventual acceptance.

Please submit your revised paper before 19-Aug-2021. Please note that the revision deadline will expire at 00.00am on this date. If we do not hear from you within this time then it will be assumed that the paper has been withdrawn. In exceptional circumstances, extensions may be possible if agreed with the Editorial Office in advance. We do not allow multiple rounds of revision so we urge you to make every effort to fully address all of the comments at this stage. If deemed necessary by the Editors, your manuscript will be sent back to one or more of the original reviewers for assessment. If the original reviewers are not available we may invite new reviewers.

Royal Society of Chemistry
Thomas Graham House
Science Park, Milton Road
Cambridge, CB4 0WF

Royal Society Open Science - Chemistry Editorial Office

On behalf of the Subject Editor Professor Anthony Stace and the Associate Editor Professor Chaohua Cui.

RSC Associate Editor:
Comments to the Author:
(There are no comments.)

RSC Subject Editor:
Comments to the Author:
(There are no comments.)

Reviewers' Comments to Author:

Reviewer: 1

Comments to the Author(s)

In this study authors explained about π -conjugated polymer and their use in metal ion batteries and supercapacitors. Authors provided the basic and known information and I am sorry to say that I can't accept this paper for publication in its present form. I am recommending authors to revise their manuscript and also do needed modification.

Table format is not good for publication and that need to modify with acceptable format.

I did not see any figures in the manuscript, of course it is reproduced with permission but for reader's it is difficult to understand the content without figures.

Reviewer: 2

Comments to the Author(s)

The manuscript provides an interesting update of pi conjugated polymers for energy storage applications. However, before publication, the following points need correction/clarification:

1. There are several grammatical errors throughout the manuscript. Those need to be corrected before the manuscript can be published.
2. Section 3.1.4 on KIBs is very unclear. A more specific elaboration of how the pi-conjugated polymers have been tried (and failed) will be beneficial for the reader.
3. The authors point out that pi-conjugated polymers have gained "enormous attention" in NIB systems as cathodes. It would be great if they could point out some notable performances in section 3.1.5.

Author's Response to Decision Letter for (RSOS-210567.R0)

See Appendix A.

RSOS-210567.R1 (Revision)

Review form: Reviewer 1

Is the manuscript scientifically sound in its present form?

Yes

Are the interpretations and conclusions justified by the results?

Yes

Is the language acceptable?

Yes

Do you have any ethical concerns with this paper?

No

Have you any concerns about statistical analyses in this paper?

No

Recommendation?

Accept as is

Comments to the Author(s)

authors revised their manuscript as per reviewer comments and i am happy to accept this paper for the publication in its present form.

Review form: Reviewer 3

Is the manuscript scientifically sound in its present form?

Yes

Are the interpretations and conclusions justified by the results?

Yes

Is the language acceptable?

Yes

Do you have any ethical concerns with this paper?

No

Have you any concerns about statistical analyses in this paper?

No

Recommendation?

Accept as is

Comments to the Author(s)

Accept

Decision letter (RSOS-210567.R1)

Dear Dr Kumar:

Title: A review of π -conjugated polymer based nanocomposites for metal-ion batteries and supercapacitors

Manuscript ID: RSOS-210567.R1

It is a pleasure to accept your manuscript in its current form for publication in Royal Society Open Science. The chemistry content of Royal Society Open Science is published in collaboration with the Royal Society of Chemistry.

Yours sincerely,
Dr Ellis Wilde
Publishing Editor, Journals

On behalf of the Subject Editor Professor Anthony Stace and the Associate Editor Professor Chaohua Cui.

RSC Associate Editor
Comments to the Author:
(There are no comments.)

RSC Subject Editor
Comments to the Author:
(There are no comments.)

Reviewer(s)' Comments to Author:

Reviewer: 1

Comments to the Author(s)

authors revised their manuscript as per reviewer comments and i am happy to accept this paper for the publication in its present form.

Reviewer: 3

Comments to the Author(s)

accept

Appendix A

Date: 10/08/2021

Dr Laura Smith

Publishing Editor, Journal,
Royal Society of Chemistry
Thomas Graham House
Science Park, Milton Road
Cambridge, CB4 0WF
Royal Society Open Science - Chemistry Editorial Office

Subject: Submission of revised manuscript ID: RSOS-210567 in "Royal Society of open Sciences".

Dear Dr Laura Smith,

Thank you very much for your email dated 27th July 2021, informing the comments on manuscript Ms. Ref. No manuscript ID: RSOS-210567 entitled, "A review of π -conjugated polymer based nanocomposites for metal-ion batteries and supercapacitors". We also thank reviewers for their valuable suggestions to our manuscript.

The manuscript is revised according to reviewer's suggestions (corrections are highlighted with yellow color). I enclose herewith the revised manuscript in the light of reviewer's report as follows:

We hope that manuscript is now in acceptable format.

Answers to comments

Reviewer #1:

We are grateful to the reviewer for valuable suggestions. As per the comments and suggestions we have now modified the manuscript. Our explanations vis-a-vis comments of the reviewer are as follows:

- 1) Table format is not good for publication and that need to modify with acceptable format

Ans: . Now, we have modified tables and arranged this in acceptable format of the journal.

- 2) I did not see any figures in the manuscript, of course it is reproduced with permission but for reader's it is difficult to understand the content without figures.

Ans: We are thankful to authors for this valuable comment. We have now introduced a explanatory figures in the revised manuscript.

Reviewer #2:

1) There are several grammatical errors throughout the manuscript. Those need to be corrected before the manuscript can be published.

Ans: , As per the reviewers comments we have thoroughly checked the manuscript for grammatical errors and for English language..

2) Section 3.1.4 on KIBs is very unclear. A more specific elaboration of how the pi-conjugated polymers have been tried (and failed) will be beneficial for the reader.

Ans: As per the comments of reviewer, the relevant discussion supported with references is included in the revised manuscript. Also, the changes made in the manuscript are highlighted with yellow color.

3) The authors point out that pi-conjugatd polymers have gained "enormous attention" in NIB systems as cathodes. It would be great if they could point out some notable performances in section 3.1.5.

Ans: As per the comments of reviewer, the relevant discussion supported with references is now included in the revised manuscript. Also, the changes made in the manuscript are highlighted in yellow color.

We hope that our revised manuscript will be accepted by the reviewer.

Yours sincerely,

Dr. Yogesh Kumar